# Learning to Enhance Low-Light Images with Reliable Attention and Reinforced Distribution Alignment

## Abstract

Low-light image enhancement (LLIE) methods have recently adopted the HVI color space, which alleviates the entanglement between luminance and color and improves color fidelity through chrominance polarization and intensity compression. However, existing approaches may suffer from error accumulation during the interaction between luminance and chrominance components, and the lack of fine-grained modeling of color distribution can lead to unsatisfactory enhancement results. To address these challenges, we propose a novel low-light image enhancement framework, Learning to Enhance Low-Light Images with Reliable Attention and Reinforced Distribution Alignment. Specifically, we introduce two key modules: the Reliable Cross Attention (RCA) module, which aggregates luminance and chrominance features with reliable queries, and the Reinforced Distribution Alignment (RDA) module, which robustly fits the color distribution in a more fine-grained manner. These designs significantly improve the quality of enhanced images under low-light conditions. Extensive experiments on multiple benchmark datasets demonstrate that our method achieves state-of-the-art performance compared with existing approaches.

## 1 Introduction

Images captured by imaging sensors in low-light conditions often suffer from significant noise. To address this issue, the task of low-light image enhancement has emerged, focusing on improving brightness, contrast, and the visibility of details in dark environments. In addition to its standalone benefits, low-light enhancement serves as an important foundation for various downstream vision tasks, such as object detection Zou et al. (2023), tracking Feichtenhofer et al. (2017), and image matching Cheng et al. (2025). However, in the conventional RGB color space, color and luminance are closely intertwined, which can lead to color distortions or unnatural brightness after enhancement. To alleviate these problems, multiple distinct categories of methods have been developed.

Traditional sRGB-based methods Wang et al. (2022a) often cause color shifts and distortions due to the coupling of luminance and chromaticity. To address this, some approaches Guo & Hu (2023) convert images to the HSV color space for more precise luminance enhancement. However, HSV introduces new issues such as red channel discontinuities and black-plane noise, resulting in visual artifacts and color distortions. To tackle these challenges, CIDNet Yan et al. (2025) introduces the HVI color space, which is specifically designed for low-light image enhancement. The HVI color space polarizes the hue-saturation plane to reduce red-channel discontinuities and employs a learnable intensity compression function to adaptively suppress low-luminance areas, effectively minimizing black noise artifacts. This design significantly enhances both color fidelity and perceived naturalness in low-light conditions. Nevertheless, residual noise and incomplete decoupling between luminance and chromaticity still introduce errors during feature interaction, which degrade the overall naturalness and smoothness of enhanced images. Moreover, due to the challenges in precisely modeling the distribution in HV space, global color distortions remain a common issue.

From our discussion on HVI-based methods, we have identified two main challenges that need to be addressed to achieve more accurate and robust low-light image enhancement. ***(1) How to efficiently aggregate luminance and chromaticity features while avoiding noise amplification.*** Previous ap-

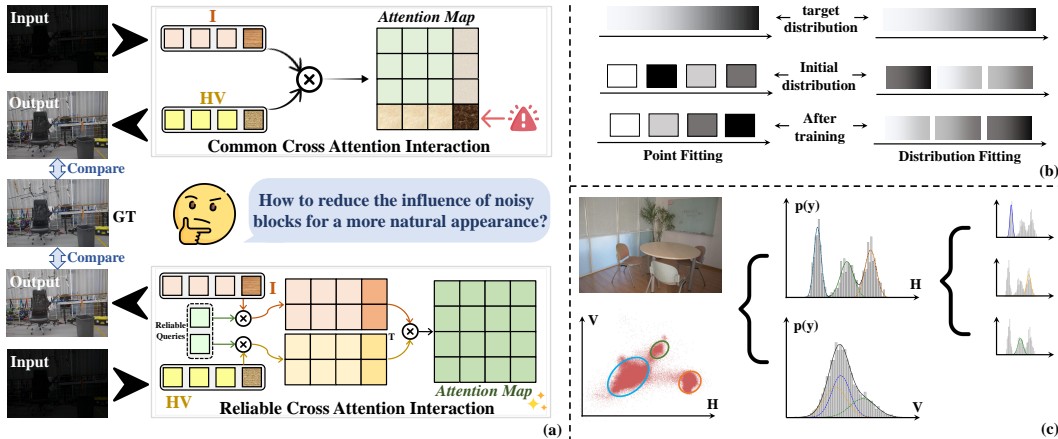

Figure 1: (a) Visualization comparing our method with traditional interactions. (b) Comparison between point-based and distribution-based modeling, showing that distribution modeling provides a more fine-grained fit to the target distribution. (c) Visualization of distribution modeling, where clustering and Gaussian mixture modeling effectively capture the distributions.

proaches often employ conventional transformer-based interactions, which tend to amplify noise when low-quality regions from luminance and color spaces are aggregated. This leads to noticeable degradation in overall image quality. To address this issue (see Figure 1(a)), we argue that leveraging high-quality features to guide aggregation can better align the enhancement process with real-scene information, thereby improving enhancement quality while suppressing the interference of noisy or invalid features in dark regions. ***(2) How to precisely constrain the distribution of chromaticity features to achieve more natural color rendering.*** Existing methods commonly adopt point-based constraints to approximate the feature distribution, but such strategies suffer from a lack of global consistency and insufficient modeling of the overall distribution (Figure 1(b)). To overcome this limitation, we propose to approximate the feature distribution with multiple fitted distributions (Figure 1(c)), enabling a more fine-grained characterization of the joint statistics between luminance and chromaticity. Furthermore, we introduce a reinforcement learning mechanism to dynamically adjust the distribution fitting process, allowing the model to adaptively select optimal distribution parameters. This not only preserves global consistency but also enhances the naturalness and robustness of the enhanced results.

To address the above challenges, we propose Learning to Enhance Low-Light Images with Reliable Attention and Reinforced Distribution Alignment, which introduces two key modules: the Reliable Cross Attention (RCA) and Reinforced Distribution Alignment (RDA). RCA employs reliable queries to bridge luminance and chromaticity features, replacing traditional transformer interactions. This design suppresses noise from redundant features, improves the naturalness of enhanced images, and reduces computational complexity for higher efficiency. RDA targets the chromaticity components (H and V), applying unsupervised clustering and Gaussian modeling to capture multiple distributions. A reinforcement learning mechanism then dynamically adjusts distribution parameters for better alignment with global chromaticity statistics. As a result, our model achieves more natural and stable enhancement, especially under low-light conditions.

In summary, our contributions are threefold:

1. We propose a novel framework, Learning to Enhance Low-Light Images with Reliable Attention and Reinforced Distribution Alignment, achieving state-of-the-art performance in low-light image enhancement.

2. We propose the Reliable Cross Attention (RCA) module to adaptively aggregate luminance and chromaticity features, suppress noise, and enhance illumination balance and color fidelity. Additionally, we introduce the Reinforced Distribution Alignment (RDA) module, which models multiple chromaticity distributions using Gaussian clustering and dynamically refines them through reinforcement learning, leading to clearer and more natural results.

3. Extensive experiments and ablation studies on ten benchmark datasets validate the superiority of our method.

## 2 RELATED WORKS

In this section, we review existing approaches for low-light image enhancement.

**Traditional Methods.** Early low-light enhancement methods were largely heuristic and did not require training data. Histogram equalization Pizer et al. (1987) and gamma correction Rahman et al. (2016) improve contrast and brightness by redistributing pixel intensities, but often ignore scene illumination, leading to over-enhanced or washed-out results. Retinex-based approaches Land & McCann (1971); Rahman et al. (2004) decompose an image into illumination and reflectance components and refine the illumination with structural priors. Although more physically motivated, they rely on idealized assumptions and are prone to noise amplification and color distortion in real-world conditions.

**Learning-Based Methods.** Deep learning has transformed low-light enhancement into a data-driven task. RetinexNet Wei et al. (2018) and KinD Zhang et al. (2019) embed Retinex decomposition into CNNs, but remain sensitive to illumination estimation, often amplifying noise or shifting colors. ZeroDCE Guo et al. (2020) and RUAS Liu et al. (2021) avoid explicit decomposition by learning pixel-adaptive curves or structural priors, but may introduce artifacts or unstable chrominance. Flow-based methods such as LLFlow Wang et al. (2022a) deliver high-fidelity restoration via normalizing flows, but incur heavy computational costs and require paired data. GAN-based approaches like EnlightenGAN Jiang et al. (2021) enhance perceptual realism through adversarial training, though sometimes at the expense of unnatural textures. UFormer Wang et al. (2022b) introduces a Transformer-based U-shaped architecture with a Locally-enhanced Window Transformer and multi-scale restoration modulator, achieving top performance in image restoration. Restormer Zamir et al. (2022a) presents an efficient Transformer model with a multi-Dconv head attention mechanism and multi-scale design, excelling in image deraining, deblurring, and denoising. MIRNet Zamir et al. (2022b) uses multi-scale residual blocks and non-local attention to preserve spatial details and context, achieving state-of-the-art results in image denoising, super-resolution, and enhancement. SNR-aware networks Xu et al. (2022) integrate noise priors to reduce artifacts but still struggle with color inconsistency. Transformer-based models, e.g., LL-Former Wang et al. (2023) and RetinexFormer Cai et al. (2023), capture long-range dependencies but lack explicit channel-level alignment. Bread Guo & Hu (2023) mitigates noise–color entanglement in YCbCr space, GSAD Hou et al. (2023) employs a global structure-aware diffusion process, and QuadPrior Wang et al. (2024) introduces physical priors to constrain illumination enhancement. However, these methods often face issues such as overexposure, color shifts, or high computational cost. RetinexMamba Bai et al. (2024) combines traditional Retinex theory with deep learning to improve illumination estimation and noise suppression for low-light enhancement. More recently, CIDNet Yan et al. (2025) explored the HVI color space to alleviate red discontinuities and black noise. Our method builds upon existing low-light image enhancement techniques, addressing key challenges in feature aggregation and chromaticity distribution modeling to achieve superior results.

## 3 METHOD

The proposed method is illustrated in Figure 2. The input image is first mapped to the HVI color space to separate luminance from chromaticity, and then passed through the Reliable Cross Attention (RCA) module. This module aggregates luminance and chromaticity features using reliable queries, effectively suppressing noise and enhancing naturalness. Next, the Reinforced Distribution Alignment (RDA) module models chromaticity distributions through unsupervised clustering and Gaussian mixture modeling, while reinforcement learning dynamically adjusts parameters for better global alignment. Finally, the enhanced HVI representation is mapped back to the RGB color space. The role of the HVI transformation is explained in the Appendix A.1.

### 3.1 RELIABLE CROSS ATTENTION MODULE

To establish robust and reliable interactions between the luminance and chromaticity features and mitigate noise amplification during cross-domain feature aggregation, we propose the Reliable Cross Attention (RCA) module. The RCA module employs a set of learnable query vectors $Q_0 \in \mathbb{R}^{M \times d}$ (where $M$ denotes the number of query vectors) initialized close to zero to extract a compact collection of high confidence descriptors denoted as $R_Q$ from the joint intensity-chromaticity representation via cross attention. The input feature maps of the intensity and chrominance branches are

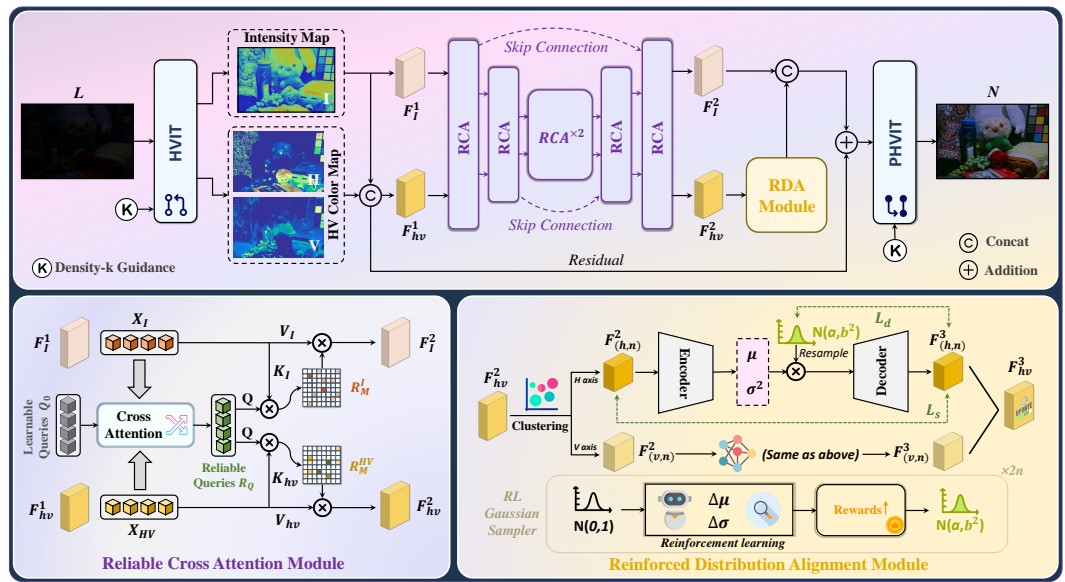

Figure 2: The overall pipeline of our method process begins by transforming the input into the HVI space. The Reliable Cross Attention (RCA) module aggregates luminance and chromaticity features with reliable queries, suppressing noise and improving naturalness. The Reinforced Distribution Alignment (RDA) module models chromaticity distributions via clustering and Gaussian modeling, with reinforcement learning dynamically refining parameters for better global alignment. The final enhanced image is reconstructed in the sRGB color space.

denoted by $F_I^1 \in \mathbb{R}^{C \times H \times W}$ and $F_{hv}^1 \in \mathbb{R}^{C \times H \times W}$, respectively, which are flattened along spatial dimensions to obtain token matrices $X_I \in \mathbb{R}^{N \times C}$ and $X_{hv} \in \mathbb{R}^{N \times C}$ with $N = H \cdot W$.

RCA operates in two stages: extracting reliable queries from the joint intensity-chromaticity representation, and guiding each branch with per-branch reliability maps that modulate the value tensors before residual fusion. In the first stage, branch-specific linear projections produce keys and values,

$$
\begin{aligned}
K_I = X_I W_K^I, \quad V_I = X_I W_V^I, \\
K_{hv} = X_{hv} W_K^{hv}, \quad V_{hv} = X_{hv} W_V^{hv},
\end{aligned}
\tag{1}
$$

where $W_K^I, W_V^I, W_K^{hv}, W_V^{hv} \in \mathbb{R}^{C \times d}$ are learned projection matrices and $d$ is the scaling factor to stabilize gradients. Through concatenation, joint key/value matrices are formed as:

$$
K_{\text{joint}} = \text{Concat}(K_I, K_{hv}) \in \mathbb{R}^{N' \times d}, \qquad V_{\text{joint}} = \text{Concat}(V_I, V_{hv}) \in \mathbb{R}^{N' \times d},
\tag{2}
$$

when $N' = 2N$. The empty queries $Q_0$ probe the joint matrix via scaled dot-product cross-attention to produce a compact set of reliable query descriptors $R \in \mathbb{R}^{M \times d}$:

$$
A_{\text{joint}} = \text{softmax}\left(\frac{Q_0 K_{\text{joint}}^\top}{\sqrt{d}}\right) \in \mathbb{R}^{M \times N'}, \qquad R_Q = A_{\text{joint}} V_{\text{joint}} \in \mathbb{R}^{M \times d}.
\tag{3}
$$

In the second stage, the reliable queries $R_Q$ act as queries against each branch key to yield per-query compatibility maps for the intensity and HV branches:

$$
R_M^I = \text{softmax}\left(\frac{R_Q K_I^\top}{\sqrt{d}}\right) \in \mathbb{R}^{M \times N}, \qquad R_M^{HV} = \text{softmax}\left(\frac{R_Q K_{hv}^\top}{\sqrt{d}}\right) \in \mathbb{R}^{M \times N}.
\tag{4}
$$

where $x \in \{1, \ldots, N\}$, $R_M^I$ and $R_M^{HV}$ denote the per-query compatibility maps in the I and HV branches, respectively. Finally, these per-branch reliability maps multiplicatively modulate the branch value tensors element-wise,

$$
\widetilde{F}_I^2 = R_M^I V_I(x), \qquad \widetilde{F}_{hv}^2 = R_M^{HV} V_{hv}(x),
\tag{5}
$$

where $F_I^2$ and $F_{hv}^2$ denote the intermediate feature representations of the intensity and HV branches, respectively. The modulated responses are projected back into the channel space and reshaped to the original spatial resolution, then fused with input features via residual addition. Subsequently, LayerNorm is applied to stabilize the feature distribution and enable reliable optimization, this step is omitted from Figure 2 for visual clarity.

Overall, the RCA module selectively aggregates reliable information across intensity and chromaticity branches while suppressing noise amplification, thereby improving enhancement quality and reducing computational complexity from $O(N^2)$ to $O(MN)$ where $M$ is significantly smaller than $N$, compared to conventional global attention mechanisms.

## 3.2 REINFORCED DISTRIBUTION ALIGNMENT MODULE

After obtaining high-quality luminance and chrominance components through the RCA module, we further consider the distribution modeling of the chrominance branch. Previous methods lack targeted constraints in this aspect, and common approaches that approximate the distribution via point-wise constraints often suffer from insufficient global consistency and inadequate modeling of overall statistics. To address this issue, we design the proposed RDA module, which models the distribution as a Gaussian Mixture Model (GMM) Rasmussen (1999) through unsupervised clustering Xie et al. (2016). Furthermore, reinforcement learning is employed to drive the policy that adaptively adjusts the posterior distribution along the $H$ and $V$ directions, thereby fitting an optimal chrominance distribution for low-light image enhancement.

Formally, given the chrominance components $F_H, F_V \in \mathbb{R}^{n \times d}$, we first perform deep embedded clustering Xie et al. (2016); Guo et al. (2018) to partition the features into $K$ groups. Each group is then parameterized by a Gaussian mixture model (GMM):

$$p(\mathbf{x}) = \sum_{k=1}^{K} \pi_k \, \mathcal{N}(\mathbf{x} \mid \mu_k, \Sigma_k), \tag{6}$$

where $\pi_k$ denotes the mixture weight, and $\mu_k, \Sigma_k$ are the mean and covariance of the $k$-th Gaussian component. This formulation allows us to capture fine-grained statistics compared to single-point constraints. Given clustered features $F_{h,n}^2$ and $F_{v,n}^2$, the encoder predicts the mean and variance $(\mu, \sigma^2)$. By adopting the reparameterization trick

$$\mathbf{z} = \mu + \sigma \odot \epsilon, \quad \epsilon \sim \mathcal{N}(a, b). \tag{7}$$

The training objective of the RDA module consists of two components: the reconstruction loss and the KL divergence regularization, which are designed to ensure visual consistency of the enhanced results and alignment of feature distributions. Given the decoder outputs $F_{h,n}^3, F_{v,n}^3$ and the original input features $F_{h,n}^2, F_{v,n}^2$, we require them to remain consistent in the feature space. Accordingly, the reconstruction loss is defined as:

$$\mathcal{L}_s = \mathbb{E}_{q_\phi(\mathbf{z}|F^2)} \big[ \|F_{h,n}^3 - F_{h,n}^2\|_2^2 + \|F_{v,n}^3 - F_{v,n}^2\|_2^2 \big], \tag{8}$$

where $q_\phi(\mathbf{z} \mid F^2)$ denotes the posterior distribution parameterized by the encoder. This term constrains the decoder to faithfully recover the input features under low-light conditions, thereby enhancing detail fidelity and luminance consistency. To prevent the posterior distribution from straying too far from the prior, which is defined by the Gaussian Mixture Model (GMM), we introduce a KL divergence regularization term:

$$\mathcal{L}_d = D_{\mathrm{KL}}\big(q_\phi(\mathbf{z} \mid F_{h,n}^2) \,\|\, p(\mathbf{z})\big) + D_{\mathrm{KL}}\big(q_\phi(\mathbf{z} \mid F_{v,n}^2) \,\|\, p(\mathbf{z})\big), \tag{9}$$

where $p(\mathbf{z})$ represents the mixture prior distribution obtained via unsupervised clustering and Gaussian modeling. This regularization encourages the encoder outputs to better align with the global statistics, effectively suppressing instability of chrominance components and avoiding undesired color shifts or distribution collapse during enhancement.

In conventional frameworks, the posterior distribution parameters $(\mu, \sigma)$ are directly predicted by the encoder and regularized to align with the prior. However, such static alignment is insufficient in low-light scenarios, where chrominance distributions exhibit large variations and noise amplification becomes severe. To overcome this limitation, we introduce a reinforcement learning (RL) mechanism into the RDA module, enabling dynamic adjustment of Gaussian parameters guided by enhancement quality feedback. We regard $(\mu, \sigma)$ as adjustable parameters and employ an RL policy network to output correction terms $(\Delta\mu, \Delta\sigma)$, yielding the updated distribution:

$$\mu' = \mu \pm \Delta\mu, \quad \sigma' = \sigma \pm \Delta\sigma, \tag{10}$$

where $(\Delta\mu, \Delta\sigma)$ are actions sampled from the policy $\pi_\theta(a \mid s)$ given the current state $s$. The state encodes the current distribution fitting quality (e.g., KL divergence and histogram statistics), allowing the policy to flexibly adapt the distribution shape under different conditions. To directly link policy optimization with enhancement performance, we design the reward $r$ as:

$$r = \frac{1}{\|F_h^3 - F_{hv,gt}^3\|_1}, \tag{11}$$

where $F_h^3$ represents the enhanced chrominance features, $F_{hv,gt}^3$ represents the ground truth chrominance features, and $\|\cdot\|_1$ denotes the L1 norm. The inverse of the $L_1$ loss encourages the policy to reduce the difference between the enhanced and ground truth features, thereby improving the enhancement quality.

We adopt the REINFORCE algorithm to optimize the policy network. The objective is defined as the expected cumulative reward:

$$\mathcal{J}(\theta) = \mathbb{E}_{a \sim \pi_\theta(\cdot|s)}[r], \tag{12}$$

where $r$ represents the reward, which is the feedback signal given for a particular action, and $\pi_\theta$ is the policy network that determines the probability distribution of actions $a$ given the state $s$. The parameter $\theta$ denotes the parameters of the policy network. The gradients of this objective are estimated as:

$$\nabla_\theta \mathcal{J}(\theta) \approx \frac{1}{N} \sum_{i=1}^{N} r_i \nabla_\theta \log \pi_\theta(a_i \mid s_i), \tag{13}$$

where $N$ denotes the number of sampled actions, $a_i$ is the action taken in the $i$-th sample, and $s_i$ is the corresponding state for the action. This optimization procedure enables the policy to iteratively refine the distribution parameters according to the feedback received from the image quality, thus enhancing the model's stability and adaptability in low-light conditions.

In order to directly link policy optimization with the enhancement performance, we design the RL loss $\mathcal{L}_r$ as:

$$\mathcal{L}_r = -\mathbb{E}_{a \sim \pi_\theta(\cdot|s)} \left[r \cdot \log \pi_\theta(a \mid s)\right], \tag{14}$$

which encourages the policy to minimize the discrepancy between the enhanced chrominance features and the ground truth, improving the enhancement quality by reducing the $L_1$ loss between the predicted and true chrominance values.

### 3.3 Loss Fuction

To constrain the training of the proposed framework, we employ a comprehensive loss that combines the primary reconstruction loss in both the RGB and HVI spaces with the VCF and CDA losses. Specifically, let $I_{\text{out}}$ and $I_{\text{gt}}$ represent the enhanced and ground-truth images in the RGB domain, and let $I_{\text{out}}^{\text{HVI}}$ and $I_{\text{gt}}^{\text{HVI}}$ represent their counterparts in the HVI color space. The reconstruction loss is defined as:

$$\mathcal{L}_t = \|I_{\text{out}} - I_{\text{gt}}\|_1 + \lambda_1 \left\| I_{\text{out}}^{\text{HVI}} - I_{\text{gt}}^{\text{HVI}} \right\|_1, \tag{15}$$

where $\|\cdot\|_1$ denotes the $\ell_1$-norm. The total loss function is then formulated as:

$$\mathcal{L} = \mathcal{L}_t + \lambda_2 \mathcal{L}_s + \lambda_3 \mathcal{L}_d + \lambda_4 \mathcal{L}_r, \tag{16}$$

where $\lambda_1$, $\lambda_2$, $\lambda_3$ and $\lambda_4$ are weighting coefficients.

## 4 Experiments

### 4.1 Datasets and Settings

**Datasets.** To validate the effectiveness of the proposed method, we conduct experiments on seven LLIE benchmark datasets, including three paired datasets: LOLv1 Wei et al. (2018), LOLv2 Yang et al. (2021), and SICE Cai et al. (2018), and four unpaired datasets, including DICM Lv et al. (2018), LIME Guo et al. (2016), MEF Ma et al. (2015), NPE Wang et al. (2013), and VV Vonikakis et al. (2018). The LOLv1 dataset has 485 paired training images and 15 for testing. LOLv2 consists of two subsets: LOLv2-Real (689 training, 100 testing) and LOLv2-Synthetic (900 training, 100 testing). The SICE dataset includes 589 paired low-light and well-exposed images, with 100 randomly selected for testing and the rest for training and validation. For SID, we convert raw images to sRGB without gamma correction, resulting in extremely dark images. We crop the training images into $256 \times 256$ patches and train for 1,000 epochs with a batch size of 4.

**Experiment Settings.** We implement the proposed method using PyTorch and train all models on a single NVIDIA RTX 3090 GPU. The optimizer is Adam Kingma & Ba (2014) with parameters $\beta_1 = 0.9$ and $\beta_2 = 0.999$. The initial learning rate is set to $1 \times 10^{-4}$ and is gradually reduced to $1 \times 10^{-7}$ using a cosine annealing schedule Loshchilov & Hutter (2016). During training, the batch size is consistently set to 8 and input images are cropped into $400 \times 400$ patches for all datasets except the LOLv2-Synthetic subset, for which full-resolution images are used without cropping. The $\lambda_1$, $\lambda_2$, $\lambda_3$ and $\lambda_4$ set to 1, 1, 0.5 and 0.5, respectively.

**Evaluation Metrics.** Following our baseline Yan et al. (2025), for paired datasets, we adopt Peak Signal-to-Noise Ratio (PSNR) and Structural Similarity Index (SSIM) Wang et al. (2004) as distortion-based metrics to evaluate reconstruction fidelity. To further assess the perceptual quality of the enhanced results, we report the Learned Perceptual Image Patch Similarity (LPIPS) Zhang et al. (2018), computed using a pretrained AlexNet Krizhevsky et al. (2012a). For *unpaired* datasets, we employ two no-reference image quality assessment metrics, BRISQUE Krizhevsky et al. (2012b) and NIQE Mittal et al. (2012), to evaluate perceptual realism. Moreover, to provide a comprehensive comparison, our method is benchmarked against 11 state-of-the-art supervised learning methods, including RetinexNet Wei et al. (2018), KinD Zhang et al. (2019), LLFlow Wang et al. (2022a), EnlightenGAN Jiang et al. (2021), SNR-Aware Xu et al. (2022), Bread Guo & Hu (2023), Pair-LIE Fu et al. (2023), LLFormer Wang et al. (2023), RetinexFormer Cai et al. (2023), GSAD Hou et al. (2023) and CIDNet Yan et al. (2025), as well as 3 unsupervised learning methods, such as ZeroDCE Guo et al. (2020), RUAS Liu et al. (2021), QuadPrior Wang et al. (2024),RetinexMamba Bai et al. (2024), UFormer Wang et al. (2022b), Restormer Zamir et al. (2022a) and MIRNet Zamir et al. (2022b) across all datasets.

Table 1: Quantitative results of PSNR↑/SSIM↑/LPIPS↓ on LOL (v1 and v2) datasets. Best performance in **purple**, second best in **cyan**.

| Methods | LOLv1 | | | LOLv2-Real | | | LOLv2-Synthetic | | |
|---|---|---|---|---|---|---|---|---|---|
| | PSNR↑ | SSIM↑ | LPIPS↓ | PSNR↑ | SSIM↑ | LPIPS↓ | PSNR↑ | SSIM↑ | LPIPS↓ |
| RetinexNet Wei et al. (2018) | 18.915 | 0.427 | 0.470 | 16.097 | 0.401 | 0.543 | 17.137 | 0.762 | 0.255 |
| KinD Zhang et al. (2019) | 23.018 | 0.843 | 0.156 | 17.544 | 0.669 | 0.375 | 18.320 | 0.796 | 0.252 |
| ZeroDCE Guo et al. (2020) | 21.880 | 0.640 | 0.335 | 16.059 | 0.580 | 0.313 | 17.712 | 0.815 | 0.169 |
| RUAS Liu et al. (2021) | 18.654 | 0.518 | 0.270 | 15.326 | 0.488 | 0.176 | 13.765 | 0.638 | 0.305 |
| EnlightenGAN Jiang et al. (2021) | 20.003 | 0.691 | 0.317 | 18.230 | 0.617 | 0.309 | 16.570 | 0.734 | 0.220 |
| LLFlow Wang et al. (2022a) | 24.998 | 0.871 | 0.117 | 17.433 | 0.831 | 0.315 | 24.870 | 0.919 | 0.067 |
| UFormerWang et al. (2022b) | 19.610 | 0.755 | 0.197 | 19.410 | 0.657 | 0.194 | 19.660 | 0.871 | 0.075 |
| RestormerZamir et al. (2022a) | 22.430 | 0.823 | 0.184 | 19.940 | 0.827 | 0.183 | 21.410 | 0.830 | 0.062 |
| MIRNetZamir et al. (2022b) | 24.140 | 0.830 | 0.154 | 20.020 | 0.820 | 0.175 | 21.940 | 0.876 | 0.058 |
| SNR-Aware Xu et al. (2022) | 26.716 | 0.851 | 0.152 | 21.480 | 0.849 | 0.163 | 24.140 | 0.928 | 0.056 |
| Bread Guo & Hu (2023) | 25.299 | 0.847 | 0.155 | 20.830 | 0.847 | 0.174 | 17.630 | 0.919 | 0.091 |
| PairLIE Fu et al. (2023) | 23.526 | 0.755 | 0.248 | 19.855 | 0.778 | 0.317 | 19.074 | 0.794 | 0.230 |
| LLFormer Wang et al. (2023) | 25.758 | 0.823 | 0.167 | 20.056 | 0.792 | 0.211 | 24.038 | 0.909 | 0.066 |
| RetinexFormer Cai et al. (2023) | 27.140 | 0.850 | 0.129 | 22.794 | 0.840 | 0.171 | 25.670 | 0.930 | 0.059 |
| GSAD Hou et al. (2023) | 27.605 | 0.876 | 0.092 | 20.153 | 0.846 | 0.113 | 24.472 | 0.929 | 0.051 |
| QuadPrior Wang et al. (2024) | 22.849 | 0.800 | 0.201 | 20.592 | 0.811 | 0.202 | 16.108 | 0.758 | 0.114 |
| RetinexMambaBai et al. (2024) | 24.030 | 0.827 | 0.146 | 22.450 | 0.844 | 0.174 | 25.890 | 0.935 | 0.054 |
| CIDNet Yan et al. (2025) | 28.201 | 0.889 | 0.079 | 24.111 | 0.871 | 0.108 | 25.705 | 0.942 | 0.045 |
| Ours | 29.123 | 0.880 | 0.075 | 24.892 | 0.875 | 0.110 | 26.512 | 0.947 | 0.040 |

## 4.2 RESULTS ON PAIRED DATASETS

We evaluate our proposed method on three widely-used low-light image enhancement benchmarks: LOLv1, LOLv2-Real, and LOLv2-Synthetic. As illustrated in Table 1, our approach consistently achieves the best performance across PSNR, SSIM, and LPIPS metrics. In contrast, existing methods exhibit notable weaknesses: RUAS and LLFlow often produce over-smoothed or distorted textures, PairLIE and GSAD suffer from unstable color rendition with visible hue shifts, while CIDNet fails to adequately suppress residual noise, resulting in unnatural tone mapping.

Table 2: Complexity of Different Methods.

| | ZeroDCE | RUAS | LLFlow | EnlightenGAN | SNR-Aware | Bread | PairLIE | LLFormer |
|---|---|---|---|---|---|---|---|---|
| **Params/M** | 0.075 | 0.003 | 17.42 | 114.35 | 4.01 | 2.02 | 0.33 | 24.55 |
| **FLOPs/G** | 4.83 | 0.83 | 358.4 | 61.01 | 26.35 | 19.85 | 20.81 | 22.52 |

Benefiting from the Reliable Cross Attention (RCA) module and Reinforced Distribution Alignment (RDA) module, our method effectively suppresses noisy or inconsistent features and ensures better alignment between luminance and chromaticity statistics. This enables more balanced illumination and natural color reproduction, particularly in extremely dark regions. Quantitatively, our approach achieves a PSNR of 29.123 on LOLv1, outperforming the previous state-of-the-art by 0.771 dB, and reaches 24.892 on LOLv2-Real with an SSIM improvement of 0.022 over the second best. On LOLv2-Synthetic, our method attains 26.512 PSNR and 0.947 SSIM, both ranking first. These results highlight the effectiveness and robustness of our design, which delivers state-of-the-art visual quality while maintaining competitive computational efficiency. Furthermore, compared with CIDNet, the proposed method introduces only a marginal increase in parameters (+0.28M) and FLOPs (+2.96G)$^2$, yet consistently outperforms it across multiple benchmark metrics, which validates the efficiency of our design and demonstrates its favorable trade-off between complexity and performance.

Figure 3 presents a qualitative comparison of the enhancement results on the LOL (v1 and v2) and SICE datasets, showcasing the performance of our method in comparison to several state-of-the-art approaches, including RUAS, LLFlow, PairLIE, GSAD, EnlightenGAN, RetinexFormer, and CIDNet. As seen in the figure, our method demonstrates superior performance in enhancing low-light images, effectively improving both brightness and contrast while preserving details in the images. Our approach outperforms the competing meth-

Table 3: Quantitative result on SID, SICE and the five unpaired datasets (DICM, LIME, MEF, NPE, and VV). The top-ranking score is in **Bold**.

| Methods | SICE | | SID | | Unpaired | |
|---|---|---|---|---|---|---|
| | PSNR↑ | SSIM↑ | PSNR↑ | SSIM↑ | BRISQUE↓ | NIQE↓ |
| RetinexNet | 12.424 | 0.613 | 15.695 | 0.395 | 23.286 | 4.558 |
| ZeroDCE | 12.452 | 0.639 | 14.087 | 0.090 | 26.343 | 4.763 |
| RUAS | 8.656 | 0.494 | 12.622 | 0.081 | 26.372 | 4.800 |
| LLFlow | 12.737 | 0.617 | 16.226 | 0.367 | 26.087 | 4.221 |
| CIDNet | 13.435 | 0.642 | 22.904 | 0.676 | 23.521 | 3.523 |
| Ours | **16.195** | **0.714** | **23.116** | **0.727** | **22.894** | **3.417** |

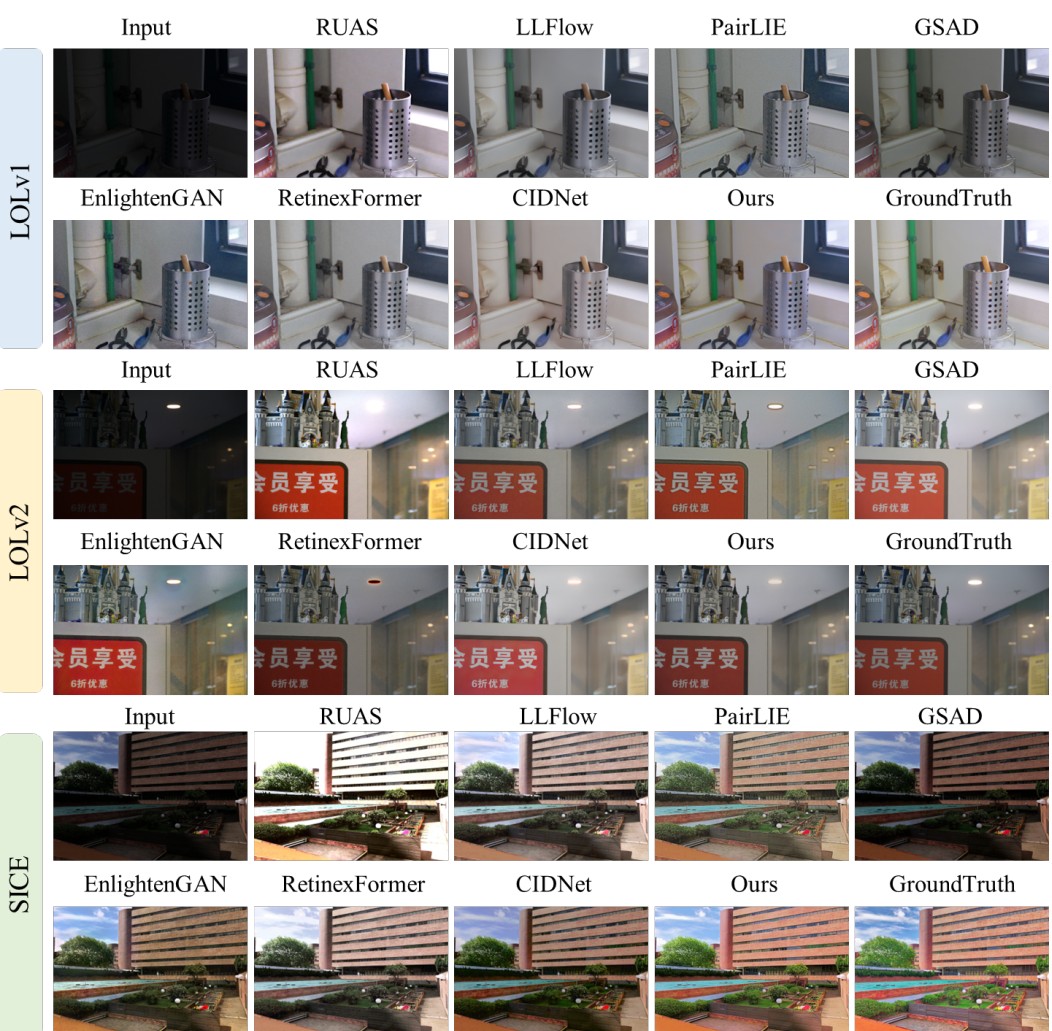

Figure 3: Qualitative comparison of enhancement results on the LOL (v1 and v2) and SICE dataset, generated by various methods.

ods in terms of naturalness and color fidelity, particularly in challenging low-light scenarios. The enhanced images show clearer details and reduced noise, providing a more visually pleasing result compared to the other methods, as highlighted in the visual comparison. The effectiveness of our method is particularly evident in the SICE dataset, where it significantly enhances the image quality without introducing noticeable artifacts.

## 4.3 RESULTS ON UNPAIRED DATASETS

We conduct comprehensive evaluations on unpaired datasets. For the unpaired datasets (DICM, LIME, MEF, NPE, and VV), we report two widely used no-reference perceptual quality measures, BRISQUE and NIQE,

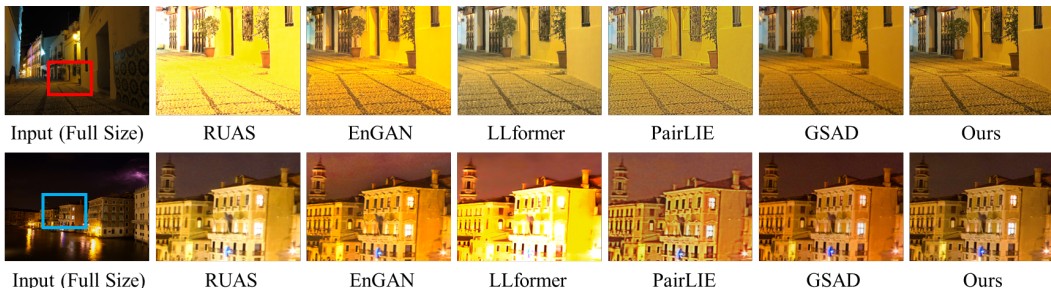

Figure 4: Qualitative comparison of enhancement results on the unpaired dataset, generated by various methods.

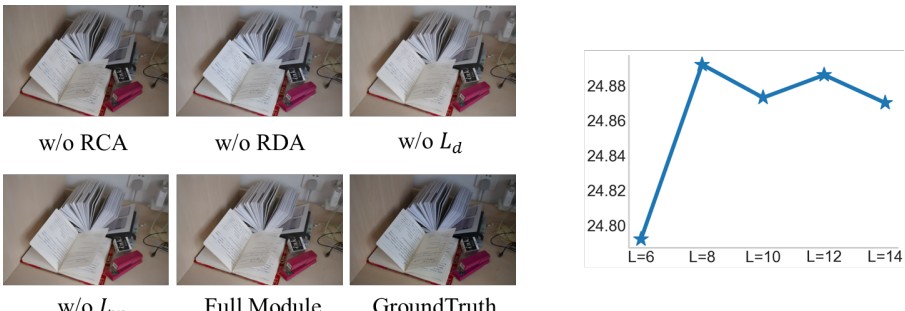

(a) Module Ablation Study        (b) Impact of the number of query vector

Figure 5: Ablation study on the LOLv2-real dataset.

to evaluate visual realism. As shown in Table 3, our method consistently achieves the best results across all metrics. Specifically, it outperforms existing supervised methods such as RetinexNet, LLFlow, and CIDNet, as well as unsupervised approaches including ZeroDCE and RUAS. On the five unpaired datasets, our model also yields the lowest BRISQUE and NIQE scores, indicating superior perceptual quality. These results demonstrate that our method not only preserves structural fidelity in paired scenarios but also generalizes effectively to real-world unpaired conditions.

## 4.4 ABLATION STUDIES

To validate the effectiveness of each component in our proposed framework, we perform a series of ablation experiments on the LOLv2-real dataset. As summarized in Table 4, the model's performance is assessed using PSNR and SSIM metrics. In the ablation studies, we sequentially remove individual modules to analyze their contributions. The removal of the RCA results in a noticeable decline in performance, underscoring the importance of adaptively aggregating luminance and chromaticity features. Additionally, excluding the RDA module or our design loss $L_d$ or $L_s$ leads to further performance degradation, highlighting the critical role of precise chromaticity distribution modeling in achieving more natural and accurate color rendering.

Table 4: Ablation studies of modules, w means with and w/o means without.

| exp. | RCA | RDA | $L_d$ | $L_s$ | PSNR↑ | SSIM↑ |
|------|-----|-----|-------|-------|-------|-------|
| 1 | w/o | w/o | w/o | w/o | 24.111 | 0.871 |
| 2 | w | w/o | w/o | w/o | 24.459 | 0.872 |
| 3 | w | w | w/o | w/o | 24.681 | 0.874 |
| 4 | w | w | w | w/o | 24.792 | 0.875 |
| 5 | w | w | w | w | 24.892 | 0.875 |

Moreover, as demonstrated in Figure 5(a), removing the RDA module causes significant degradation in color fidelity, as it eliminates the mechanism for aligning chromaticity distributions. The absence of the RCA module leads to substantial deterioration in both luminance and chrominance quality, highlighting its critical role in suppressing noise and maintaining balanced illumination and accurate color restoration during enhancement.

In the ablation study shown in Figure 5(b), we evaluate the impact of the number of query vectors, $L$, on performance. Increasing $L$ from 6 to 8 leads to significant improvement, with diminishing returns as $L$ exceeds 8. This suggests that $L = 8$ offers a good balance between performance and computational efficiency.

## 4.5 CONCLUSION

In this work, we introduce a novel low-light image enhancement framework, Learning to Enhance Low-Light Images with Reliable Attention and Reinforced Distribution Alignment, which effectively addresses the challenges of noise amplification and color distortion in low-light conditions. Our framework combines the Reliable Cross Attention (RCA) module and the Reinforced Distribution Alignment (RDA) module to improve the interaction between luminance and chromaticity features while preserving color fidelity and naturalness. The RCA module suppresses redundant features and enhances the efficiency of feature aggregation, while the RDA module refines chromaticity distributions through unsupervised clustering and reinforcement learning. Extensive experiments on ten benchmark datasets demonstrate that our method achieves state-of-the-art performance, offering superior visual quality and strong generalization across various lighting conditions. Our results confirm that VCR is an effective and efficient solution for low-light image enhancement, pushing the boundaries of image quality under challenging conditions.

ETHICS STATEMENT

This research adheres to the ICLR Code of Ethics. We ensure that no ethical violations have occurred during the research process. All datasets used comply with publicly available privacy policies, and we have ensured the security and privacy of the data during collection and use. There are no conflicts of interest or funding issues in this research. All methods and applications used in this research follow principles of fairness and objectivity to ensure the integrity and transparency of the research.

REPRODUCIBILITY STATEMENT

All improvements in this research are based on open-source code and datasets. We provide comprehensive experimental details and algorithm descriptions, including the models, datasets, and training processes used. All relevant source code and datasets will be made open-source. We encourage readers to use the same experimental setups and parameters to reproduce our results and validate the theories and algorithms presented in this work, ensuring the reproducibility of the research and supporting the validation of the results.

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

# A  APPENDIX

## A.1  HVI COLOR SPACE

In the standard sRGB color space, image brightness and chromatic information are tightly coupled across the three color channels, which may disrupt the perceived illumination or color balance of the entire image when making adjustments to any individual channel. Although the HSV color space separates intensity from chromaticity, it inadvertently amplifies noise in regions of extreme red and near-black areas, producing pronounced "red-discontinuity" and "black-plane" artifacts during enhancement. To address the above limitations, the HVI color space has been proposed to alleviate inherent color noise, which is composed of three channels: $I_{\max}$, $\hat{H}$, and $\hat{V}$, designed to mitigate the artifacts introduced by the HSV representation. Here, $C_k(x)$ denotes a learnable intensity collapse function that remaps the maximum intensity $I_{\max}(x)$ for stabilizing low-light responses. The parameter $k$, termed *density-k*, controls the density of black-plane points in HVT/PHVIT, thereby balancing noise suppression and detail preservation. According to the Max-RGB, for each individual pixel $x$, the intensity map of image $I$ can be estimated:

$$I_{\max}(x) = \max_{c \in \{R,G,B\}} I_c(x). \tag{17}$$

Meanwhile, according to the sRGB-HSV transformation, the saturation $s$ of the image can be obtained:

$$s = \begin{cases} 0, & I_{\max} = 0 \\ \dfrac{I_{\max} - \min(I_c)}{I_{\max}}, & I_{\max} \neq 0 \end{cases} \tag{18}$$

and the hue $h$ of the image is formulated as follows:

$$h = \begin{cases} 0, & \text{if} \quad s = 0 \\ \left( \dfrac{I_G - I_B}{I_{\max} - \min(I_c)} \right) \bmod 6, & \text{if} \quad I_{\max} = I_R \\ 2 + \dfrac{I_B - I_R}{I_{\max} - \min(I_c)}, & \text{if} \quad I_{\max} = I_G \\ 4 + \dfrac{I_R - I_G}{I_{\max} - \min(I_c)}, & \text{if} \quad I_{\max} = I_B \end{cases} \tag{19}$$

where $s$ and $h$ correspond to any pixel in the saturation map $S(x)$ and hue map $H(x)$, respectively. Moreover, corresponding to HVT in Figure 2, the horizontal chromaticity component $\hat{H}(x)$ and the vertical component $\hat{V}(x)$ are constructed by polarizing the hue angle from HSV into Cartesian space, defined as:

$$\begin{cases} \hat{H}(x) = C_k(x) \cdot S(x) \cdot \cos\left( \dfrac{\pi H(x)}{3} \right), \\ \hat{V}(x) = C_k(x) \cdot S(x) \cdot \sin\left( \dfrac{\pi H(x)}{3} \right), \end{cases} \tag{20}$$

where $C_k(x)$ is a learnable intensity collapse function defined as:

$$C_k(x) = k \cdot \sqrt{\sin\left(\frac{\pi I_{\max}(x)}{2}\right) + \varepsilon},\tag{21}$$

with $k$ as a trainable parameter and $\varepsilon$ as a small constant (set to $10^{-8}$) for training stability. Moreover, as shown in Fig. 2, the Perceptual-inverse HVI Transformation (PHVIT) is performed to convert the HVI space back to HSV. The hue $H(x)$, saturation $S(x)$, and value $V(x)$ are estimated as:

$$\begin{cases} H(x) = \dfrac{1}{2\pi} \cdot \arctan\left(\dfrac{\hat{v}(x)}{\hat{h}(x)}\right) \bmod 1, \\ S(x) = \alpha_S \cdot \sqrt{\hat{h}^2(x) + \hat{v}^2(x)}, \\ V(x) = \alpha_I \cdot I_{\max}(x), \end{cases}\tag{22}$$

where $\alpha_S$ and $\alpha_I$ are linear scaling parameters that control the output image's saturation and brightness, respectively. The normalized intermediate chromaticity coordinates are computed as:

$$\begin{cases} \hat{h}(x) = \dfrac{\hat{H}(x)}{C_k(x) + \varepsilon}, \\ \hat{v}(x) = \dfrac{\hat{V}(x)}{C_k(x) + \varepsilon}. \end{cases}\tag{23}$$

## A.2 BLIND/REFERENCELESS IMAGE SPATIAL QUALITY EVALUATOR (BRISQUE)

BRISQUE Krizhevsky et al. (2012b) is a blind image quality assessment method that leverages natural scene statistics (NSS) in the spatial domain. It normalizes local luminance values and characterizes their distribution through an asymmetric generalized Gaussian distribution (AGGD). From this distribution, descriptive statistics such as shape and variance are derived. A support vector regression model, trained with subjective quality annotations, maps these statistics to a perceptual quality score. Lower BRISQUE values correspond to higher visual quality. Since it operates without reference images, BRISQUE is particularly suitable for evaluating real-world or unpaired data.

## A.3 NATURALNESS IMAGE QUALITY EVALUATOR (NIQE)

NIQE Mittal et al. (2012) is another no-reference quality assessment approach, designed to capture deviations from the statistical regularities of natural images. It first constructs a multivariate Gaussian model using NSS-based features (e.g., mean-subtracted contrast-normalized coefficients and local pixel correlations) extracted from pristine natural images. For a test image, the same features are computed, and the Mahalanobis distance to the Gaussian model is used as the quality score:

$$\mathrm{NIQE}(I) = \sqrt{(f - \mu)^\top \Sigma^{-1} (f - \mu)},$$

where $f$ represents the feature vector of the test image, and $\mu, \Sigma$ denote the mean and covariance estimated from natural data. A lower NIQE value implies stronger alignment with natural image statistics and thus better perceptual quality.

## A.4 FAILURE CASES

Figure 6 presents typical failure cases on the unpaired dataset (DICM). In extreme scenarios, our method occasionally struggles with inadequate brightness restoration and insufficient noise suppression. We plan to

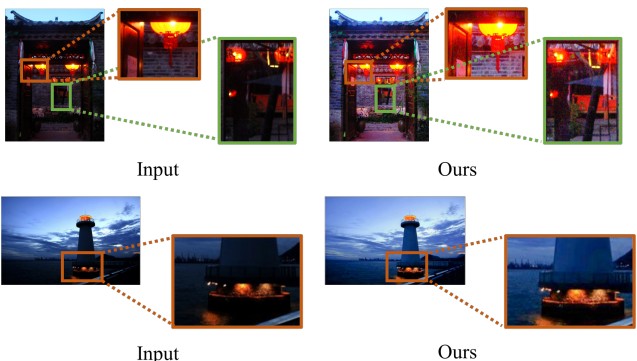

Figure 6: Failure cases.

address these limitations in future work by integrating explicit noise priors, enforcing temporal consistency in video sequences, and introducing locally adaptive color temperature adjustment.

## A.5 USE OF LARGE MODELS

In this work, large language models are employed solely for language polishing and improving the readability of the manuscript. They are not involved in problem formulation, algorithm design, model implementation, or experimental analysis. All technical contributions and experimental results are independently developed and verified by the authors.

