# OpenReview forum: "Learning to Enhance Low-Light Images with Reliable Attention and Reinforced Distribution Alignment"
_ICLR.cc/2026/Conference — ICLR 2026 Conference Withdrawn Submission_

### Official Review · Reviewer_vWub · 2025-10-21

**Soundness:** 2
**Presentation:** 1
**Contribution:** 2
**Rating:** 4
**Confidence:** 4

**Summary:**

This paper proposes a novel framework for low-light image enhancement, titled Learning to Enhance Low-Light Images with Reliable Attention and Reinforced Distribution Alignment. The method introduces two key modules: Reliable Cross Attention (RCA), which effectively fuses luminance and chrominance features using reliable queries, and Reinforced Distribution Alignment (RDA), which models color distributions in a fine-grained and robust manner. Experimental results across multiple benchmark datasets show that the proposed approach achieves state-of-the-art performance.

**Strengths:**

- The proposed RCA module effectively reduces computational complexity by utilizing reliable queries for efficient fusion of luminance and chrominance features.

- The method achieves state-of-the-art performance across multiple benchmark datasets, demonstrating strong effectiveness and generalization capability.

**Weaknesses:**

- The experimental results do not sufficiently support the main claims of the proposed method.
- In the introduction section, it is unclear how the authors substantiate the claim that the feature interaction process leads to noise amplification. An additional analysis or empirical evidence would strengthen this point.
- The rationale for applying a reinforcement learning algorithm to constrain the distribution of chromatic features is not clearly justified, and the effectiveness of the reinforcement learning loss (L_r) is not demonstrated in the ablation studies.
- In Table 2, the parameters (M) and FLOPs (G) of the proposed method are not reported, making it difficult to assess the model’s computational efficiency.

**Questions:**

Please refer to Weaknesses.

---

### Official Review · Reviewer_ve9J · 2025-10-31

**Soundness:** 2
**Presentation:** 2
**Contribution:** 2
**Rating:** 2
**Confidence:** 5

**Summary:**

To reduce the error accumulation during the interaction between luminance and chrominance components, this paper proposes a LLIE framework with Reliable Attention and Reinforced Distribution Alignment. In Reliable Attention, a learnable queriers Q is leveraged to produce high-quality queries, which is used for later cross-attention calculation. As for Reinforced Distribution Alignment, the chromaticity distributions is modeled via clustering and Gaussian modeling, and the posterior distribution parameters ($\mu$, $\sigma$) are adjusted iteratively by a learned policy net $\pi_\theta(a|s)$. Experiments are conducted on serveral LLIE benchamark datasets and a few unpaired real-world datasets.

**Strengths:**

1. The idea of employing RL algorithm for chrominance adjustment is promising.
2. The first three sections (Abstract + Sections 1-3) are well-organized and easy to follow.

**Weaknesses:**

1. Figure 1 is not clear enough. For example, what's the advantages of the distribution fitting compared to point fitting? Is there any possibility that the fitted distribution is continuous, similar to the target distribution? Besides, In (c), what's the last diagram for? I guess it's the gaussian of mixtures, but it's not clear sufficiently.

2. The overview for learning-based methods is not comprehensive, especially lacking many papers (Flow-based: LLFlow+SKF; Codebook-based: RQ-LLIE, GLARE; Retinex-based: ECMamba) published in top venues.

3. As the core of the Reliable Attention Module, the idea of learnable query vectors is first proposed in PromptIR and then is used in many later works. The originality of this part is a concern to me.

4. Some details in the methods are not clear to me. (1) The calculation of $R_Q$ is not presented in Figure 2; (2) If $M <<N$, the illustration of  $R_M^{I}$ and  $R_M^{HV}$ are impossible to be square; (3) The downscaling and upscaling of SCA are not discussed in the manuscript; (4) Are a and b hyperparameters in equation (7)

5. The presentation of experiments is poor. (1) For SICE, why are test images randomly selected? It's better to follow baselines to split train/test subsets; (2) I have concerns on numbers reported in Table 1: results of some methods are calculate using GT-mean (LLFlow, Retinexformer), while some are calculated without GT-mean (RetinexMamba). Besides, I believe many baseline results are copied from their  original paper or obtained from their official weights, however, the training configurations of this paper (training on 400*400 patches) are different baselines, leading to a risk of performance bias; (3) Table 2 seems to be imcomplete, no results for CIDNet and the proposed method. As this method includes a RL process, I also suggest the authors to provide inference latency comparisons.

6. The letter size in pages 8-9 is smaller than table/figure captions, I am not sure if this is a formatting issue. For the $L_w$ and $L$ shown in Figure 5, I can not find any definitions for these variables.

**Questions:**

Please see weakness for my questions.

---

### Official Review · Reviewer_66AK · 2025-10-31

**Soundness:** 2
**Presentation:** 3
**Contribution:** 1
**Rating:** 2
**Confidence:** 5

**Summary:**

This manuscript proposes a novel low-light image enhancement framework by introducing a reliable cross-attention and reinforced distribution alignment. Comprehensive experiments and ablation analyses across ten benchmark datasets consistently validate the superiority of the proposed method.

**Strengths:**

1. The manuscript is well written, with clear logic and readability.
2. This manuscript proposes a Reliable Cross Attention (RCA) module that adaptively fuses luminance and chromaticity features, effectively suppressing noise while improving illumination balance and color fidelity.
3. This manuscript introduces a Reinforced Distribution Alignment (RDA) module, which models multiple chromaticity distributions via Gaussian clustering and progressively refines them through reinforcement learning, thereby producing clearer and more natural visual results.

**Weaknesses:**

1. The innovation is limited, and the overall algorithm framework is similar to CIDNet. More importantly, the core Reliable Cross Attention Module is very similar to the LCA in CIDNet.
2. Figure 5(a) does not match the corresponding content in Table 4. Does L_w correspond to L_s?
3. The ablation study analysis in section 4.4 does not match the content presented in Table 4. When RCA is removed, RDA, L_d, and L_s are also removed. This does not directly evaluate the effectiveness of RCA.

**Questions:**

1. The innovation is limited, and the overall algorithm framework is similar to CIDNet. More importantly, the core Reliable Cross Attention Module is very similar to the LCA in CIDNet.
2. Figure 5(a) does not match the corresponding content in Table 4. Does L_w correspond to L_s?
3. The ablation study analysis in section 4.4 does not match the content presented in Table 4. When RCA is removed, RDA, L_d, and L_s are also removed. This does not directly evaluate the effectiveness of RCA.

---

### Official Review · Reviewer_gUGS · 2025-11-01

**Soundness:** 2
**Presentation:** 3
**Contribution:** 2
**Rating:** 4
**Confidence:** 4

**Summary:**

This paper conducts low-light image enhancement in the HVI color space. It analyzes the problems of error accumulation that occur during the interaction between brightness and chromaticity components in current methods, as well as the lack of fine-grained modeling of color distribution, which leads to suboptimal enhancement results. To address these issues, the study proposes the RCA module and the RDA module, and validates their effectiveness through experiments on multiple datasets.

**Strengths:**

This paper improves upon the shortcomings of existing methods based on the HVI color space. It introduces an RCA module that enables a flexible, cross-color-space, spatial-aware attention mechanism while reducing its complexity to ensure efficiency. In addition, a novel RDA module is proposed to model chromaticity and adjust it using reinforcement algorithm, enhancing the global color consistency and ensuring adequate modeling of overall statistics.

**Weaknesses:**

1. As mentioned in your contributions and Chapter 3, the use of RCA can suppress noise amplification during the cross-domain feature aggregation process. However, your paper does not provide evidence to support this claim. There are no experiments to demonstrate its denoising capability, and the visual ablation results presented do not clearly show its noise suppression effect. The discussion in the paper only proves that RCA is a flexible cross-domain information interaction module, but it cannot substantiate that it mitigates noise amplification.
2. About the RDA module, the paper states: “In conventional frameworks, the posterior distribution parameters are directly predicted by the encoder and regularized to align with the prior. However, such static alignment is insufficient in low-light scenarios, where chrominance distributions exhibit large variations and noise amplification becomes severe.” However, the paper does not provide any evidence or discussion to support this claim, nor does it include ablation experiments showing the results under conventional frameworks. Therefore, the argument lacks sufficient persuasiveness. The RDA module combines clustering, VAE, and further introduces RL. Given such a complex structural design, you need to justify the necessity of incorporating RL.
3. Table 2 in the paper compares the number of parameters and FLOPs, but the values for the proposed method itself are missing.

**Questions:**

Please refer to Weaknesses.

---

### Note · Authors · 2025-11-16

I have read and agree with the venue's withdrawal policy on behalf of myself and my co-authors.